# Catalytic Pyrolysis of Sawdust with Desulfurized Fly Ash for Pyrolysis Gas Upgrading

**DOI:** 10.3390/ijerph192315755

**Published:** 2022-11-26

**Authors:** Jinling Song, Chuyang Tang, Xinyuan An, Yi Wang, Shankun Zhou, Chunhong Huang

**Affiliations:** School of Civil Engineering, University of Science and Technology Liaoning, Qianshan Road, Anshan 114051, China

**Keywords:** pyrolysis, catalytic, sawdust, desulfurized fly ash, pyrolysis gas upgrading

## Abstract

In this study, the catalytic effects of desulfurized fly ash (DFA) on the gaseous products of sawdust (SD) pyrolysis were investigated in a tubular furnace. The results indicated that DFA catalyzed the process of SD decomposition to improve the hydrogen content and the calorific value of pyrolysis gas. As to its effect on pyrolysis products, DFA increased the non-oxide content of CH_4_, C_3_H_4_, and H_2_ in pyrolysis gas by 1.4-, 1.8-, and 2.3-fold, respectively. Meanwhile, the catalytic effect of DFA reduced the CO and CO_2_ yields during DFA/SD pyrolysis. Based on the model compound method, CaSO_3_ and Ca(OH)_2_ in DFA was proved to have quite different catalytic effects on pyrolysis gas components. Ca(OH)_2_ accelerated the formation of CH_4_ and H_2_ through the cracking of methoxyl during lignin and cellulose degradation, while CaSO_3_ favored the generation of CO and CO_2_ due to the carbonyl and carboxyl of lignin in SD. CaSO_3_ also catalyzed SD pyrolysis to promote the C_3_H_4_ yield in pyrolysis gas. Overall, the catalytic pyrolysis of SD with DFA yielded negative-carbon emission, which upgraded the quality of the pyrolysis gas.

## 1. Introduction

In recent decades, the availability of biomass as a carbon-neutral energy source has drawn more concerns than ever due to the challenges of greenhouse gas emissions and environmental pollution [1,2]. Pyrolysis is one of the promising technologies in biomass conversion research [3]. Low-temperature pyrolysis is a type of thermochemical process occurring under 600 °C that converts biomass into pyrolysis gas, pyrolytic water, bio-oil, and semi-coke [4]. In general, pyrolysis gas and bio-oil derived from the process often possess a high oxygen content and low calorific value [5,6]. As a result, the main purpose of the catalysts in the pyrolysis process focused on increasing the yield and improving the quality of the target products [7]. Many studies documented the catalyst of alkaline soil considerably improving the production and quality of pyrolysis gas during biomass pyrolysis [8,9]. Bingbing Qiu et al. [10] revealed that AAEMs (alkali and alkaline soil metals) significantly promoted the decomposition of biomass and boosted the volatile fraction yield. Moreover, AAEMs accelerated the pyrolysis gas generation and decreased the semi-coke yield during biomass gasification, according to a study by Lei Deng et al. [11]. AAEMs as catalysts also regulated the biomass pyrolysis process to increase the hydrogen yield [12]. Hao Song et al. [13] further summarized the key factors affecting hydrogen production from biomass pyrolysis. It indicated that calcium-based catalysts had a significant impact on increasing the hydrogen yield and inhibiting the formation of CO_2_. However, catalysts can rarely be effectively recovered during the catalytic pyrolysis of biomass [14,15]. As a result, AAEMs-loaded catalysts were limited in the application of the efficient conversion of biomass.

Desulfurized fly ash (DFA) is a byproduct derived from the tail gas purification of coal burning, which is composed of about 85% CaSO_3_ and Ca(OH)_2_ [16]. China generated 130 million tons of DFA in 2019. Most of the DFA was converted into low-value desulfurized gypsum [17,18]. Therefore, a large amount of DFA is currently deposited in the open air, which has seriously harmed the environment [19]. DFA as a type of industrial solid waste, with small particles and a large surface area [20]. However, CaSO_3_ can decompose at temperatures above 650 °C [21], which hampers the use of DFA as a catalyst in the technical method of biomass gasification [22].

In this study, DFA was utilized for the catalytic pyrolysis of SD below 600 °C. Pyrolysis experiments were used to investigate the impact of DFA additives on the distribution of pyrolytic products. The effect of DFA catalysis on the yield and quality of pyrolysis gas derived from SD pyrolysis was further analyzed by model compounds.

## 2. Materials and Methods

### 2.1. Materials and Characterization

In this study, desulfurized fly ash (DFA) was obtained from Anshan Steel II power generation, and wood chips (SD) were collected from fast-growing poplar in Liaoning Province, China. The samples of SD and DFA were dried at 105 °C for 8 h and then crushed to less than 0.178 mm. The cellulose content of SD (23.68% ± 0.05) was determined by the Van Soest method [23]. Microcrystalline cellulose (MCC) with a particle size of 20–100 μm and a polymerization of 3000–10,000 glucose units was chosen as a model compound for the pyrolysis experiments. Anhydrous calcium sulfite (AR grade) was also selected as a model compound. Ultimate analyses were determined using a Macro Cube (Elementar; Berlin, Germany). The proximate and ultimate analyses of SD and MCC are shown in Table 1.

Approximately 4 g of the DFA sample was fed into an X-ray fluorescence spectrometer for qualitative and semi-quantitative analysis. The angular reproducibility of the Shimadzu XRF-1800 equipment was 0.0001°, and the analytical diameter was ≥500 μm. The accuracy of the elemental content determination was 0.0001 wt.%. The XRF results for DFA are listed in Table 2. It is shown that the DFA has a high Ca and S content, with 58.54% and 21.24% of its total weight occurring in the form of CaO and SO_3_, respectively. In addition, the DFA sample also contains small quantities of Fe, Mg, Si, Al, Na, and K.

### 2.2. Pyrolysis Experiment

The catalytic pyrolysis experiments for SD and DFA were performed in a tube furnace, as illustrated in Figure 1. The experimental method was introduced in our previous work [24]. SD and DFA were mixed according to the mass ratio as the test samples, and about 7 g of the mixed specimens were weighed and placed in a quartz tube reactor. Next, the specimens were heated from room temperature to 600 °C at 5 °C/min and then held for 15 min. The reaction tube and liquid phase condensation collector were weighed again after the reactor was cooled. The difference between the weight of the quartz tube reactor and the condensation collector before and after the experiment was then calculated to obtain the yield of pyrolysis semi-coke and pyrolysis liquid phase products, respectively. Pyrolysis gas was collected using a gas bag, and its volume (V) was finally determined using the drainage method. The relative content (Si) of each component (Vi) in the pyrolysis gas was determined by gas chromatography (GC). The calculated method of each component is shown in Equation (1), where i is component of pyrolysis gas.
Vi = Si · V(1)

### 2.3. Thermogravimetric Analysis

A TG-DTA/DSC (Setaram, France) thermogravimetric analyzer was used to investigate the pyrolysis characteristics of SD and DFA. About 30 mg of the samples were taken into the thermogravimetric analyzer. Then, the sample was placed in a thermogravimetric balance. High purity nitrogen was introduced as a protective gas at a purging rate of 20 mL/min. The sample was heated from room temperature of 30 °C to 900 °C at a temperature increase rate of 10 °C/min. The experimental results are shown in Figure 2.

### 2.4. Gas Chromatography Experiment

The pyrolysis gas components were detected using a GC 126 (INESA Instrument Co., Shanghai, China) gas chromatography mass spectrometer, equipped with a flame ionization detector and a thermal conductivity detector. The pyrolysis gas derived from the pyrolysis experiment was passed to GC 126 at a rate of 30–50 mL/min to determine the main components (H_2_, CH_4_, CO, CO_2_, and C2−C4 hydrocarbons).

## 3. Results

### 3.1. TG Analysis of SD and DFA

The thermal weight loss (TG) and thermal weight loss rate (DTG) curves of SD and DFA are illustrated in Figure 2. The thermal decomposition of SD consists of three stages, in accordance with temperature variation. The SD sample loses free water in the first stage, from room temperature to 140 °C. The second stage is within the scope of 140 °C–410 °C, and comprises two periods. The DTG of the initial period (140 °C–210 °C) varies slightly. When the temperature exceeds 220 °C, the DTG of SD grows rapidly. The maximum DTG is 0.900%/min in this stage. When the temperature exceeds 410 °C, the SD sample initiates carbonization in the third stage. The rate of sample mass loss slows down and the temperature influence declines due to weight loss. After the temperature exceeds 550 °C, the thermal decomposition of SD is basically complete. As shown in Figure 2, the DTG curve of DFA consists of four stages. The release of extraneous water in DFA and the removal of crystalline water from the dehydration reaction of CaSO_3_–0.5H_2_O occurs in the first stage (<200 °C). This results in a weight loss of 2.5% of the DFA sample. In the second stage, the TG is about 7.5% due to the decomposition reaction of Ca(OH)_2_ in DFA. The maximum DTG of the second stage is 0.0704%/min and eventuates at 400 °C. Because of CaCO_3_ decomposition to release CO_2_, the thermal decomposition of DFA presents the maximal DTG (0.163%/min, 660 °C), and the TG is 14.59% in the third stage (530 °C–740 °C). In the fourth stage (>740 °C), the weight loss is mainly attributed to the SO_2_ emission generated from the decomposition of CaSO_3_.

### 3.2. Product Distribution from DFA/SD Pyrolysis

Figure 3 presents the product distribution of catalytic pyrolysis at different DFA/SD ratios. With the DFA content increase in the blend, the bio-oil yield decreases significantly at first, and then increase slowly. The minimum yield of bio-oil is obtained at the 50/100 ratio, with 23.43 wt.% (daf). The bio-oil yield derived from SD pyrolysis is 34.57 wt.% (daf), which is higher than that of DFA/SD pyrolysis.

Compared with the varied yields of bio-oil, the water yield increases first, and then decreases during catalytic pyrolysis. The maximum water yield (21.55 wt.%, daf) occurred at the DFA/SD ratio of 50/100. Moreover, the semi-coke yield range of catalytic pyrolysis is from 30.5 wt.% (daf) to 33.88 wt.% (daf), which if very close to that of SD pyrolysis. The maximum value of the pyrolysis gas yield during DFA/SD catalytic pyrolysis was 25.29 wt.% (daf) at the blending ratio of 50/100. When DFA/SD is more than 50/100, the DFA catalysis inhibits the formation of pyrolysis gas and leads to a decrease in the pyrolysis gas yield.

As shown in Figure 4, the pyrolysis gas content derived from the catalytic pyrolysis of DFA/SD and SD pyrolysis mainly includes CO, CH_4_, CO_2_, C_3_H_4_ (i.e., allene), C_2_H_6_, H_2_, and C_3_H_8_. DFA as a catalyst varies the yields of pyrolysis gas during SD/DFA pyrolysis. The pyrolysis gas yield generated from catalytic pyrolysis increases first and then decreases with the increasing DFA content in the sample. The minimum pyrolysis gas yield is 166.67 (mL/g, daf), which occurred at the DFA/SD ratio of 40/100. In both the SD pyrolysis and the catalytic pyrolysis of DFA/SD, the CO yield is higher than any component yields in the pyrolysis gas. The CO yield from DFA/SD pyrolysis is lower than that of SD pyrolysis. Compared with the CO yield of SD pyrolysis (32.36 mL/g, daf), DFA as a catalyst notably inhibited the CO formation during DFA/SD pyrolysis, as shown in Figure 4. Moreover, the CO yield increased with the increase in DFA content in the samples during DFA/SD pyrolysis. The CO yield at the DFA/SD ratio of 40/100 is 24.67 mL/g (daf) and declines to 23.76% that obtained by SD pyrolysis. The CO_2_ yield at the 40/100 ratio is just 61.48 mL/g (daf) and as much as 86.77% that obtained by SD pyrolysis. The CH_4_ yield of DFA/SD pyrolysis obtained the maximum, which was 38.99% more than that obtained by SD pyrolysis. In addition, the catalytic pyrolysis of DFA/SD achieves the C_3_H_4_ yield (9.52 mL/g, daf) and H_2_ yield (1.39 mL/g, daf) at the 40/100 ratio, which is 82.03% and 230.95% more than that obtained by SD pyrolysis, respectively.

Figure 5 shows the variation in pyrolysis gas yields and calorific values for different DAF/SD ratios in the catalytic pyrolysis. Based on the calculated elemental content of C, H, and O derived from the GC analysis of pyrolysis gas in Figure 5a, the result suggests that the O content decreased with the increase in DFA content in the DFA/SD blend, while the H content (0.40 mol/L) and C content (0.19 mol/L) achieved their maximum values at the mixing ratio of 40/100, respectively. Moreover, the H content at the 40/100 ratio is 73.91% more than that in the SD pyrolysis gas. With the escalation of the DFA/SD ratio, the H/C of pyrolysis gas continues to increase, and the O/C ratio declines accordingly. As shown in Figure 5b, the greatest calorific value of DAF/SD pyrolysis gas is 11.08 MJ/m^3^, obtained at the ratio of 40/100, which is 64.88% more than that obtained by SD pyrolysis.

### 3.3. Effects of Material Components on Pyrolysis Gas Products

Figure 6a presents the product distribution of CaSO_3_/SD, DFA/SD, and DFA/MCC pyrolysis. Because the cellulose content of SD is 23.68 wt.%, the DFA/MCC ratios in accordance with the DFA/SD ratios are 10/23.68, 20/23.68, 30/23.68, 40/23.68, 50/23.68, 60/23.68, 80/23.68, and 23.68/100. Based on the XRF results of DFA, the content of CaSO_3_ is 31.86 wt.%. Therefore, the corresponding ratios of CaSO_3_/SD blending are 3.19/100, 6.37/100, 9.56/100, 12.74/100, 15.93/100, 19.12/100, 25.49/100, and 31.86/100, respectively. As shown in Figure 6a, DFA as a catalyst promoted the semi-coke yields of DFA/SD pyrolysis in the range of 20/100–100/100 ratio. Moreover, the semi-coke yield of DFA/MCC pyrolysis continually improved with the increase in DFA content in the range of 10/100–60/100. On the contrary, the semi-coke yield decreased with the increase in CaSO_3_ content in the CaSO_3_/SD blends.

As shown in Figure 6b, the bio-oil yields of DFA/MCC pyrolysis are higher than those of DFA/SD and CaSO3/SD pyrolysis. The bio-oil yields of CaSO_3_/SD pyrolysis are higher than those of DFA/SD pyrolysis in the range of the 20/100–100/100 ratios. As seen in Figure 6c, the pyrolysis gas yields of DFA/SD and CaSO_3_ are prominently higher than those of DFA/MCC. On the other hand, the pyrolysis gas yields of DFA/SD are consistently greater than that those of CaSO_3_/SD pyrolysis in the ratio range of 10/100–50/100. Figure 6d reveals that the main compounds in DAF and SD cause the different effects on the water formation during catalytic pyrolysis. The DFA catalyzes the SD decomposition and promotes the yield of pyrolysis water in the process of DFA/SD pyrolysis. As the DFA content in the sample rises from 10/100 to 40/100, the pyrolytic water yield increases from 14 wt.% (daf) to 22 wt.% (daf), accordingly. In contrast, DFA significantly inhibited the generation of pyrolytic water derived from the MCC decomposition during DFA/MCC pyrolysis. The water yield of DFA/MCC pyrolysis decreased with the increase in DFA content in the samples, particularly in the 10/100–40/100 ratio range. The water yield from DAF/SD pyrolysis was consistently greater than that of the DAF/MCC ratio in the range of 30/100 to100/100, while, the water yield of DAF/SD pyrolysis was more than that of CaSO_3_/SD pyrolysis within the ratio scope of 30/100–60/100.

Figure 7 presents the component variation of pyrolysis gas derived from the pyrolysis of CaSO_3_/SD, DFA/SD, and DFA/MCC at different blending ratios. The results imply that CaSO_3_ in DAF and cellulose in SD have obvious effects on the gaseous component distribution of DAF/SD pyrolysis by means of the model compound method. As seen in Figure 7a and b, the orders of CO and CO_2_ yields can be described as CaSO_3_/SD > DFA/SD > DFA/MCC. Moreover, the CO and CO_2_ yields of CaSO_3_/SD pyrolysis are more than double and treble those of DFA/SD pyrolysis, respectively. This indicates that CaSO_3_ in DFA catalyzes DFA/SD pyrolysis and significantly improves the formation of CO and CO_2_. The great difference in the CO_2_ yield between CaSO_3_/SD and DFA/SD pyrolysis also verifies that CO_2_ generated from SD decomposition is absorbed by the Ca(OH)_2_ contained in DFA during DFA/SD pyrolysis [25]. In comparing the CO and CO_2_ yields from DFA/SD and DAF/MCC pyrolysis, it is further noted that CO and CO_2_ mainly generate from the carbonyl and carboxyl of lignin in SD [26]. As shown in Figure 7c, CaSO_3_ in DFA play a major role in promoting the C_3_H_4_ formation from SD pyrolysis. In addition, the C_3_H_4_ yields of DFA/MCC are more than those of DFA/SD in the range of the experimental ratios. This suggests that C_3_H_4_ is mainly produced by the catalytic effect of CaSO_3_ in DFA during DFA/SD pyrolysis.

As showed in Figure 7d, the CH_4_ yields of DFA/SD pyrolysis are much higher than those of CaSO_3_/SD and DFA/MCC pyrolysis. The results demonstrate that Ca(OH)_2_ in DFA accelerates the CH_4_ formation derived from the cracking of methoxyl from lignin during DFA/SD pyrolysis. CaSO_3_/SD pyrolysis scarcely produces any H_2_, which is shown in Figure 7e, because the H_2_ yield of SD pyrolysis is 0.42 mL/g, which is much lower than that of DFA/SD pyrolysis, and Ca(OH)_2_ in DFA catalyzes the generation of hydrogen during DFA/SD pyrolysis.

## 4. Discussion

The product yields of DFA/SD pyrolysis denote that as a catalyst, DFA inhibits the generation of bio-oil and increases the water yield during DFA/SD pyrolysis, as shown in Figure 4. Nevertheless, the semi-coke yield of catalytic pyrolysis was less affected by the DFA ratio in the sample. The pyrolysis gas yield generated from DFA/SD pyrolysis was less influenced by the DFA catalysis when the DFA content was less than 40/100. Nevertheless, DFA additives promoted the generation of CH_4_ during catalytic pyrolysis. The DFA catalysis also restrained the generation of CO_2_ during catalytic pyrolysis. Furthermore, DFA exhibited a significant catalysis in accelerating the formation of C_3_H_4_, C_2_H_6_, and H_2_ in pyrolysis gas. In brief, the results showed that DFA catalyzed the processing of pyrolysis gas formation during DFA/SD catalytic pyrolysis. The DFA catalysis significantly decreased the CO and CO_2_ content in pyrolysis gas and improved the yields of CH_4_, C_3_H_4_, and H_2_ in the meantime.

The element contents and calorific values of pyrolysis gas indicated that DFA catalyzes the processing of DFA/SD pyrolysis, as shown in Figure 5. Furthermore, it was worth noticing that the calorific value positively correlated with the hydrogen content derived from the pyrolysis gas of catalytic pyrolysis. Contrastingly, the oxygen content was inversely proportional to the hydrogen content. This revealed that the dehydrogenation of DFA improves the calorific value and quality of the pyrolysis gas during DFA/SD pyrolysis.

The results shown in Figure 6 indicate that CaSO_3_ has a significant catalytic effect of promoting the thermal decomposition of SD, and the Ca(OH)_2_ in DFA might contribute to the formation of coke during SD pyrolysis. Based on the variation in the semi-coke yield from DFA/SD pyrolysis, the cellulose in SD was easily converted into semi-coke by the catalytic effect of DFA. The thermal decomposition of cellulose in SD was the main source producing bio-oil during DFA/SD pyrolysis. Moreover, the bio-oil yield of DFA/MCC continued to grow with the increase in the DFA ratio in the blend. The result further indicated that DFA exerted effects on cellulose in promoting the yield of bio-oil. CaSO_3_ showed better catalytic effects than DAF in catalyzing the thermal decomposition of SD due to the production of bio-oil. Compared with the water yield of SD pyrolysis (14 wt.%, daf), CaSO_3_ accelerated water formation during SD pyrolysis. Xinyu Lu et al. [27] reported that Ca(OH)_2_ contributed markedly to the formation of pyrolysis water during lignocellulose pyrolysis. Therefore, this result further suggested that Ca(OH)_2_, as the second-most prevalent component in DFA, also promoted the SD decomposition to improve pyrolytic water yield during DFA/SD pyrolysis. Furthermore, cellulose in SD would be the main source of major pyrolytic water production during DFA/SD pyrolysis.

The catalytic effect of DFA was superior to that of CaSO_3_, and it produced more pyrolysis gas during DFA/SD pyrolysis. Furthermore, other components in DFA, such as Ca(OH)_2_, also promoted the formation of pyrolysis gas derived from DFA/SD pyrolysis. Since SD mainly consisted of cellulose, hemicellulose, and lignin, this suggested that pyrolysis gas was primarily generated by the decomposition of lignin during the DFA/SD pyrolysis. Moreover, CaSO_3_ in DFA contributed to pyrolysis gas formation from SD pyrolysis. Furthermore, the H_2_ yields of DFA/MCC pyrolysis are 2.5 times higher than those of DFA/SD. This showed that the major yield of H_2_ is produced by cellulose degradation during DFA/SD pyrolysis [28].

## 5. Conclusions

The catalytic pyrolysis of SD with DFA at different blend ratios was performed in a tubular furnace. Compared with the pyrolysis gas of SD pyrolysis, DFA as a catalyst increased the hydrogen component and the low calorific value of the pyrolysis gas from DFA/SD pyrolysis by 2.8- and 1.6-fold, respectively. DFA regulated the thermal decomposition of SD to increase the non-oxide content in pyrolysis gas (CH_4_, C_3_H_4_ and H_2_). Further investigation showed that Ca(OH)_2_ in DFA enhanced the cracking of methoxyl from lignin and cellulose degradation to promote CH_4_ and H_2_ yields. The results suggested that the DFA catalyst upgraded the quality of pyrolysis gas. On the other hand, CaSO_3_ favored the generation of C_3_H_4_ during the pyrolysis of DFA and SD. Meanwhile, the amount of carbon emissions can be reduced by the decrease in CO and CO_2_ in pyrolysis by DFA. Therefore, SD pyrolysis with DFA, at a ratio of 40/100, was the more optimal condition for negative-carbon emissions and the upgrading of pyrolysis gas.

## Figures and Tables

**Figure 1 ijerph-19-15755-f001:**
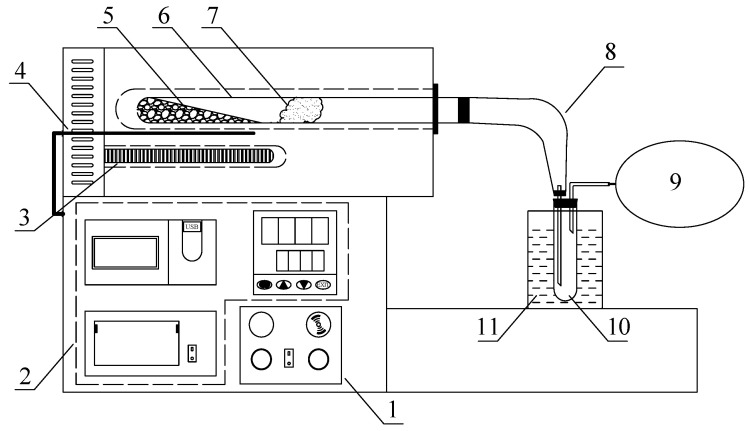
Schematic diagram of the pyrolysis apparatus. 1. heating control; 2. temperature regulation; 3. bolt electric heaters; 4. thermocouple; 5. samples; 6. tubular reactor; 7. asbestos wool; 8. plain bend; 9. gas bag; 10. condenser; 11. ice bath.

**Figure 2 ijerph-19-15755-f002:**
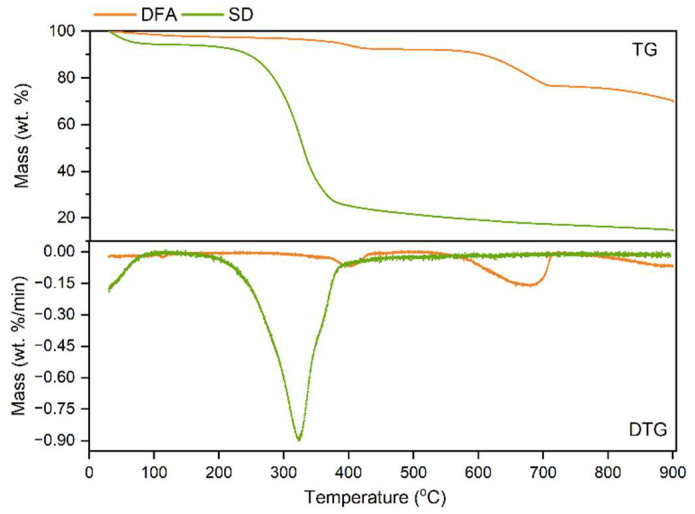
TGA and DTG curves of SD and DFA.

**Figure 3 ijerph-19-15755-f003:**
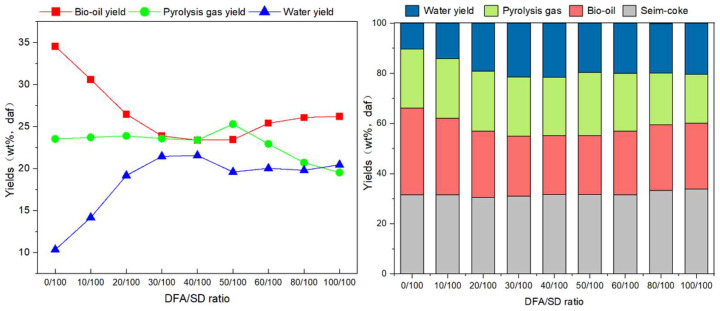
The product yields from catalytic pyrolysis of DFA and SD.

**Figure 4 ijerph-19-15755-f004:**
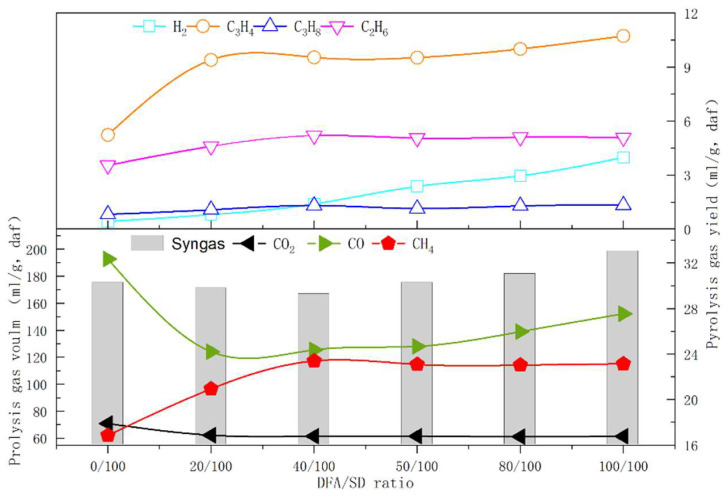
GC analyses of pyrolysis gas derived from pyrolysis.

**Figure 5 ijerph-19-15755-f005:**
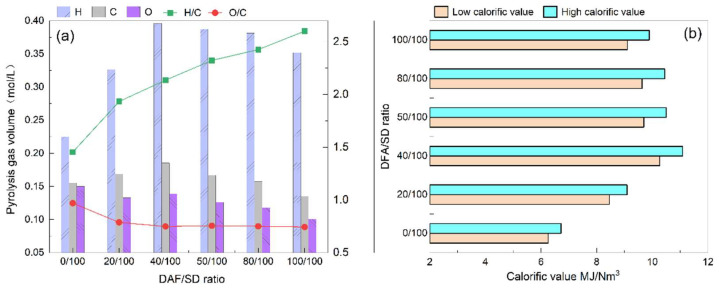
Element contents and calorific values of pyrolysis gas: (**a**) variation of C, H, and O contents in pyrolysis gas; (**b**) calorific values of pyrolysis gas.

**Figure 6 ijerph-19-15755-f006:**
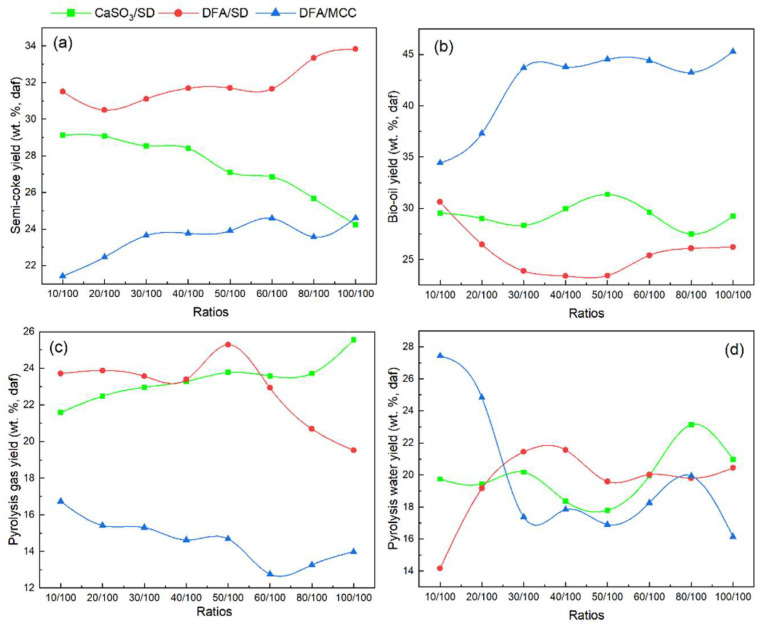
Production distribution from the pyrolysis of CaSO_3_/SD, DFA/SD, and DFA/MCC: (**a**) semi-coke yield; (**b**) bio-oil yield; (**c**) pyrolysis gas yield; (**d**) pyrolysis water yield.

**Figure 7 ijerph-19-15755-f007:**
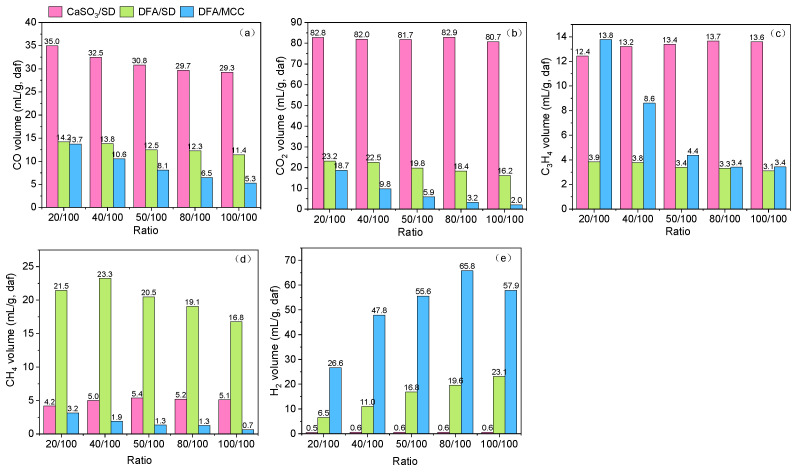
Gaseous components derived from GC results of pyrolysis gas: (**a**) CO yield; (**b**) CO_2_ yield; (**c**) C_3_H_4_ yield; (**d**) CH_4_ yield; (**e**) H_2_ yield.

**Table 1 ijerph-19-15755-t001:** Proximate and ultimate analysis results of SD.

Proximate Analysis (wt.%)		Ultimate Analysis (wt%, daf)	
Moisture (ad ^1^)	4.22	C	48.44
Ash (d ^1^)	0.75	H	6.14
Volatile Matter (daf ^1^)	75.85	N	0.69
Fixed Carbon (daf)	19.18	S	0.09
--	--	O ^2^	44.64

^1^ ad = air dried basis; d = dry basis; daf = dry and ash free basis. ^2^ Calculated by difference.

**Table 2 ijerph-19-15755-t002:** XRF results of DFA (wt.%).

Sample	Content (%)	Sample	Content (%)
CaO	58.54	Cl	1.65
SO_3_	21.24	F	1.20
SiO_2_	7.77	TiO_2_	0.36
Al_2_O_3_	3.80	K_2_O	0.31
MgO	2.66	Na_2_O	0.18
Fe_2_O_3_	2.16	--	--

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
