# Peer review of "Catalytic Pyrolysis of Sawdust with Desulfurized Fly Ash for Pyrolysis Gas Upgrading"

_ijerph, 2022, doi:10.3390/ijerph192315755_

Round 1

Reviewer 1 Report

The manuscript investigated the  catalytic effects pf desulfurization fly ash (DFA) on the  pyrolysis of sawdust.  The topic is interesting to the readers, the work is sufficient and the results are reliable.  I think it can be accepted after minor revision. 

1), The English is not fluent. For example:  The results indicated that DFA cata lyzed the process of SD decomposition to improve the hydrogen content and the calorific value of syngas significantly.

2) The figures can be presented with better quality. 

Author Response

Dear reviewer,

We are very grateful for your letter and Reviewers’ comments on our manuscript of [ijerph-2011827]. According the comments, we carefully revised the manuscript and used the track change function to make changes. In this letter, on behalf of the authors, I would like to answer Reviewer’s comments point by point.

Reviewer's comments and suggestions:

The manuscript investigated the catalytic effects pf desulfurization fly ash (DFA) on the pyrolysis of sawdust. The topic is interesting to the readers, the work is sufficient and the results are reliable. I think it can be accepted after minor revision.

1     The English is not fluent. For example:  The results indicated that DFA catalyzed the process of SD decomposition to improve the hydrogen content and the calorific value of syngas significantly.

Answer: Thank you for your suggestion. Based on your suggestion, we have revised this sentence already as follows:

The results indicated that DFA catalyzed SD pyrolysis to increase the hydrogen content and the calorific value of pyrolysis gas.

2     The figures can be presented with better quality.

Answer: Thanks for your professional suggestions on our manuscript. We have revised the problems related to the numerical expressions involved in this paper.

We would like to take this opportunity to thank you for all your time involved and this great opportunity for us to improve the manuscript. We also appreciate your clear and detailed feedback and hope that the explanation has fully addressed all of your concerns. If you have any question on the revisions, please don’t hesitate to let me know.

Sincerely yours,

Chuyang Tang

School of Civil Engineering, University of Science and Technology Liaoning, 185#, Qianshan Road, Liaoning Province 114051, PR China

E-mail: astcy@126.com

Reviewer 2 Report

Tang and co-workers have tested desulfurization ash (DFA) as catalyst in pyrolysis of sawdust. They have shown that the yield of syngas increased using DFA at a DFA/SD ratio of 50%. They have also compared these results to pyrolysis of sawdust using CaSO3 and, pyrolysis of microcrystallline cellulose using DFA. These additional experiments helped in understanding the effect of DFA on sawdust pyrolysis yields. The background research and references are cited very well. In my opinion, this research is interesting and publishable in this journal. However, I have some concerns that should be addressed before.

Comments related to research:

1. In Figure 7c (component variation of syngas derived from pyrolysis) why there is no change in the yield of C3H4 with increase in DFA loading? Increase in DFA should mean increase in CaSO3 as well and therefore, should have increased the yield?

2. In Figure 6c, why the yield of syngas using DFA/SD is higher only at 50/100 ratio and shown to decrease below or higher this ratio? Did the authors repeat this experiment to ensure the results seen?

General Comments:

I understand that the authors are not native english speakers so there can be errors. Minor comments related to language of the paper:

1. Line 48 - 51 "In brief, DFA has the...gasification". These sentences are not clear and could be rewritten for clear undesrtanding of the readers. For example, 

[However, CaSO3 can decompose at temperatures above 650 0C which hamper the use of DFA as catalyst in technical method of biomass gasification.]

2. There is unnecessary repeatedly use of the word "obviously" and can be removed. In many cases the the results were not that obvious.

3. Please rephrase this sentence in the conclusions section, Line 290-292 "Meanwhile, the catalytic....pyroysis." (This sentence is not clear)

Author Response

Dear reviewer,

We are very grateful for your letter and Reviewers’ comments on our manuscript of [ijerph-2011827]. According the comments, we carefully revised the manuscript and used the track change function to make changes. In this letter, on behalf of the authors, I would like to answer Reviewer’s comments point by point in Arial font.

Reviewer's comments and suggestions:

Tang and co-workers have tested desulfurization ash (DFA) as catalyst in pyrolysis of sawdust. They have shown that the yield of syngas increased using DFA at a DFA/SD ratio of 50%. They have also compared these results to pyrolysis of sawdust using CaSO3 and, pyrolysis of microcrystalline cellulose using DFA. These additional experiments helped in understanding the effect of DFA on sawdust pyrolysis yields. The background research and references are cited very well. In my opinion, this research is interesting and publishable in this journal. However, I have some concerns that should be addressed before.

Comment 1: In Figure 7c (component variation of syngas derived from pyrolysis) why there is no change in the yield of C3H4 with increase in DFA loading? Increase in DFA should mean increase in CaSO3 as well and therefore, should have increased the yield?

Answer: Thanks for your comments. Because the aromatic structure in the sawdust will not be broken at 600oC, C3H4 in the pyrolysis gas would mainly come from the decomposition of cellulose units. As shown in Figure 7 (c), CaSO3 improved C3H4 formation during CaSO3/SD. Moreover, the C3H4 yields of DFA/ MCC were more than that of DFA/SD at the range of 20/100 – 100/100. The CH4 yields of DFA/SD pyrolysis were significantly higher than that of DFA/MCC and CaSO3/SD pyrolysis. There was a competitive relationship between methane and butylene formation during SD pyrolysis. Moreover, CaO in DFA may inhibit the C3H4 formation. Further conclusions will be proved by more characterization analysis.

Comment 2: In Figure 6c, why the yield of syngas using DFA/SD is higher only at 50/100 ratio and shown to decrease below or higher this ratio? Did the authors repeat this experiment to ensure the results seen?

Answer: We greatly appreciate your careful review. DFA acts as a catalyst to promote the generation of CH4 and H2. At the same time, CaO in DFA obviously inhibits the formation of CO and CO2 by absorbing CO2 and H2O generated from SD pyrolysis. As a result, the maximum yield of pyrolysis gas was occurred at 50/100. The same experiment was repeated three times. And the relative error was less than 5%.

I understand that the authors are not native English speakers so there can be errors. Minor comments related to language of the paper:

  1. Line 48 - 51 "In brief, DFA has the...gasification". These sentences are not clear and could be rewritten for clear understanding of the readers. For example,

[However, CaSO3 can decompose at temperatures above 650℃ which hamper the use of DFA as catalyst in technical method of biomass gasification.]

Answer: Thank you for your advices. I have revised the manuscript according to your suggestion for the reader's clear understanding. (Line 48-51)

  1. There is unnecessary repeatedly use of the word "obviously" and can be removed. In many cases the results were not that obvious.

Answer: Thank you for your reminder. I have adjusted the manuscript to remove the unnecessary "obviously" cases based on your advice.

  1. Please rephrase this sentence in the conclusions section, Line 290-292 "Meanwhile, the catalytic...pyrolysis." (This sentence is not clear)

Answer: Thank you for your careful review. I have revised the sentence based on your suggestion as follows:

Meanwhile, the amount of carbon emissions can be reduced by the decreasing of CO and CO2 in pyrolysis by DFA. (Line 290-292)

Sincerely yours,

Chuyang Tang

School of Civil Engineering, University of Science and Technology Liaoning, 185#, Qianshan Road, Liaoning Province 114051, PR China

E-mail: astcy@126.com

Reviewer 3 Report

The manuscript „Catalytic Pyrolysis of Sawdust with Desulfurization Fly Ash for Syngas Upgrading“ is quite well written, informative and it contains lots of relevant and comprehensive literature background. Nevertheless, there is a minor stylistic issue that should be addressed by the authors before publishing:

Lines 11-12 and 286-284: Does “As to the effect on pyrolysis products, DFA increased the non-oxide content of CH4, C3H4 and H2 in syngas by 38.99%, 82.03% and 230.95%, respectively.” mean that “As to the effect on pyrolysis products, DFA increased the content of CH4, C3H4 and H2 in syngas 1.4, 1.8 and 2.3 fold, respectively.” ? If yes, “175 %” should be also removed from the conclusions. Mostly, 0-100 % is considered as rigorous values but nothing more than 100% shell be referred in “%”.

Therefore, minor revisions were suggested.

Author Response

Dear reviewer,

We are very grateful for your letter and Reviewers’ comments on our manuscript of [ijerph-2011827]. According the comments, we carefully revised the manuscript and used the track change function to make changes. In this letter, on behalf of the authors, I would like to answer Reviewer’s comments point by point.

Reviewer's comments and suggestions:

Lines 11-12 and 286-284: Does “As to the effect on pyrolysis products, DFA increased the non-oxide content of CH4, C3H4 and H2 in syngas by 38.99%, 82.03% and 230.95%, respectively.” mean that “As to the effect on pyrolysis products, DFA increased the content of CH4, C3H4 and H2 in syngas 1.4, 1.8 and 2.3fold, respectively.”? If yes, “175 %” should be also removed from the conclusions. Mostly, 0-100 % is considered as rigorous values but nothing more than 100% shell be referred in “%”.

Answer: Thank you for your comments. Based on your suggestion, replace the increases of 38.99%, 82.03% and 230.95% with increases of 1.4, 1.8 and 2.3 fold, respectively. (Lines 11-12) At your suggestion, change the increases from 175% and 63.69% to 2.8 and 1.6 fold, respectively. (Lines 289-290)

We would like to take this opportunity to thank you for all your time involved and this great opportunity for us to improve the manuscript. We also appreciate your clear and detailed feedback and hope that the explanation has fully addressed all of your concerns. If you have any question on the revisions, please don’t hesitate to let me know.

Sincerely yours,

Chuyang Tang

School of Civil Engineering, University of Science and Technology Liaoning, 185#, Qianshan Road, Liaoning Province 114051, PR China

E-mail: astcy@126.com
